# Occupational Infections among Dental Health Workers in Germany—14-Year Time Trends

**DOI:** 10.3390/ijerph181910128

**Published:** 2021-09-27

**Authors:** Rebecca Malsam, Albert Nienhaus

**Affiliations:** 1Competence Center for Epidemiology and Health Services Research for Healthcare Professionals (CVcare), Institute for Health Services Research in Dermatology and Nursing (IVDP), University Medical Center Hamburg-Eppendorf (UKE), 20246 Hamburg, Germany; rebecca.malsam@gmail.com; 2Department for Occupational Medicine, Hazardous Substances and Health Sciences (AGG), Institution for Statutory Accident Insurance in the Health and Welfare Services (BGW), 22089 Hamburg, Germany

**Keywords:** dental health workers, dentist, COVID-19, hepatitis, tuberculosis, occupational health, occupational disease

## Abstract

Dental health workers (DHW) are at increased risk of acquiring occupational infections. Due to various protective measures, it can be assumed that infections have decreased over the past 14 years. Secondary data from a German accident insurance company was analyzed in terms of reported and confirmed occupational diseases (OD) in DHW from 2006 to 2019. A total of 271 claims were reported, of which 112 were confirmed as OD, representing an average of eight per year. However, the number of claims and confirmed ODs has decreased by 65.6% and 85.7%, respectively. The decrease was most evident for hepatitis B (HBV) and C (HCV) infections, while tuberculosis (TB) infections were stable. A total of 44 HCV, 33 HBV, 6 TB and 24 latent TB infections were confirmed as ODs. For DHW, 0.05, and for hospital workers, 0.48 claims per 1000 full-time equivalents (FTE) were registered in 2019. In a separate documentation system, between March 2020 and February 2021, 155 COVID-19 claims were registered, and 47 cases were confirmed as ODs. For DHW, 0.7, and for hospital workers, 47.3 COVID-19 claims per 1000 FTE were registered since 2020. Occupational infectious diseases rarely occur among DHW. Nevertheless, new infectious diseases such as COVID-19 pose a major challenge for DHW. Continued attention should be paid to infectious disease prevention.

## 1. Introduction

Dental health workers (DHW) carry an occupational risk of exposure to several pathogens due to the special features of dental treatment. Although occupational infections and their prevention have always been an issue for health workers, they are experiencing increasing attention during the current coronavirus disease 2019 (COVID-19) pandemic. In March 2020, The New York Times published data from the U.S. Department of Labor stating that dental hygienists had the highest occupational risk of contracting severe acute respiratory syndrome coronavirus 2 (SARS-CoV-2) among all occupations analyzed, followed closely by dentists. This high risk is due to exposure to pathogens and physical proximity to patients on a daily basis [1]. Of particular interest for DHW is aerosol transmission of the virus [2,3], since aerosols frequently arise during dental interventions. The use of ultrasonic scalers, high-speed air rotors, air-water syringes and air polishing generates splatter and aerosols [4,5] that pose a risk of infection with air-borne pathogens [6]. Under experimental conditions, SARS-CoV-2 remains detectable in the environment for up to three hours after an aerosol-generating procedure [7].

Dentistry in Italy was largely affected by COVID-19 and service was reduced to emergencies [8]. Fortunately, another study from Italy reported that aerosol spreading can be reduced by technical means, e.g., the use of saliva ejectors [9]. Additional airing (e.g., opening windows) seems to be counterproductive in this setting, following the same authors. Similar results were observed in a study from the U.S. Low copy numbers of the SARS-CoV-2 virus were found in the saliva of several asymptomatic patients but none in aerosols generated from these patients. When standard infection control measures are used, dental treatment seems not to be a factor in increasing the risk for transmission of SARS-CoV-2 in asymptomatic patients [10].

In addition to the novel SARS-CoV-2, other pathogens such as influenza viruses [11] or M. tuberculosis [12] may also be transmitted through the air, posing another risk of infection for DHW.

In addition to the abovementioned pathogens, another possible risk of infection is posed by blood-borne viruses. DHW routinely deal with syringe needles and sharp instruments like burs, scalers, scalpels and endodontic files, which makes the risk of injury particularly high [13,14,15,16]. In Europe, 72% of dentists sustain at least one percutaneous injury during their professional lives, mostly caused by burs, followed by needles [13]. These percutaneous injuries and mucocutaneous contact with blood carry the risk of infecting DHW with blood-borne pathogens such as hepatitis B and C (HVB, HCV) and human immunodeficiency virus (HIV) [17,18].

In Europe, the number of infections caused by percutaneous injuries among healthcare workers (HCW) each year is estimated to be 290 cases for HCV, 210 for HBV, and 6 for HIV [19]. Cleveland and colleagues describe case reports of transmissions between patients and to DHW due to failures in complying with health protection guidelines in the U.S. [20]. Following Nagao et al. in Japan, the rate of hepatitis virus infection among dentists is higher than that among other healthcare workers due to increased exposure to both saliva and blood. Screening of DHW is therefore recommended by the authors [21]. For the U.K., an increased risk of HCV infection in DHW is described by Lodi et al. [22]. In addition to the personal burden of these diseases, they cause high costs for the healthcare system. In Germany, between 2000 and 2014, a total of EUR 88 million was spent on medical treatment and compensation for the inability to work due to occupationally acquired HCV infections among HCW [23].

Protecting DHW as a vulnerable professional group should be a priority from a public health perspective, both ethically and economically. The U.S. Needlestick Safety and Prevention Act, which was passed in 2000 and contained measures for the prevention of needlestick injuries (NSI), was decisive here (Public Law 106–430). Accordingly, employers are required to provide sharps with safety devices when available; additional requirements for a sharps injury protocol have also been added. The European Union followed this example in 2010 by implementing the Framework Agreement on prevention from sharps injuries [24]. Measures that must be implemented by the employer include provision of safety-engineered sharps devices, if available, and safer sharps disposal. Employers must also train HCW in safer working practices. Furthermore, it is necessary for DHW to wear personal protective equipment (PPE) such as medical masks, goggles and gloves during treatment, for infection prevention [24,25]. To minimize the distribution of splatter and aerosols, the use of rubber dams and high-volume evacuators is recommended [26,27]. The use of rubber dams can reduce the amount of aerosols and thus reduce the bacterial contamination of the environment by up to 70% [27]. Employers are obliged to provide employees with PPE and protective vaccinations if they are at increased risk of infection. However, vaccination against blood-borne viruses is currently only available for the hepatitis B virus. As part of the regular occupational health examination, vaccination status is checked and refreshed if necessary, at the expense of the employer [28].

Due to these measures and decreasing infection rates in the general population in Europe, it can be assumed that the incidence of infection in the dental setting has decreased. For example, a decrease of legionella infection in DHW since 1996 was described in a literature review by Petti and Vitalli in 2017 [29]. In Germany, hygiene management in dental practices improved between 2002 and 2009 after guidelines for infection prevention in dentistry were issued by the Robert Koch Institute (RKI) [30]. However, to our knowledge, no data is available on long-term time trends of infections in DHW. Therefore, the purpose of this study is to analyze time trends on occupationally acquired infections among DHW in Germany.

## 2. Materials and Methods

This analysis is based on the routine data set of the Institution for Statutory Accident Insurance and Prevention in the Health and Welfare Services (BGW—Berufsgenossenschaft für Gesundheitswesen und Wohlfahrtspflege). The BGW is an accident insurance company for non-governmental healthcare facilities in Germany that insures about nine million employees, of whom 409,427 work in dental settings, including dentists, dental assistants, dental prophylaxis assistants, dental hygienists and dental technicians.

If there is reasonable suspicion that an occupational infectious disease is present, this must be reported to the accident insurance institutions. A reasonable suspicion of OD, and thus an obligation to report, exists when exposure-prone procedures are performed and there is evidence of infection. Contact with infectious patients or materials is not reportable. The suspicion can be reported by physicians, health insurers, employers or the insured person. If an employer reports several infections in the course of an outbreak, a claim is documented for every worker concerned. For an infection to be confirmed as an OD, it must be proven that the insured person has an increased risk of infection. This is the case, for example, in surgical activities with blood contact. Furthermore, there must be a plausible connection between the exposure (injury, blood contact) and the disease itself. The accident insurer then initiates an examination procedure to determine whether an OD as defined by the Occupational Disease Ordinance and thus a claim for compensation exists.

In this study, reportable infectious ODs of DHW from 2006 to 2019 are taken into account and compared with those in hospital workers, general practice workers and specialist practice workers. The number of full-time equivalents (FTE) was used to calculate the claim rate per 1000 FTE. The FTE were obtained from BGW. The FTEs are an estimate of the number of workers covered by the compensation board. For example, two half-time workers add up to one FTE. In logistic regression, the odds for a claim by DHW were compared to the odds for hospital workers, general practice workers and specialist practice workers. Besides odds ratios (OR), 95%-confidence intervals were calculated. In addition, tests for trend were performed. A *p*-value < 0.5 was considered statistically significant. The reference year was 2006.

Two data sets were used for this analysis; the standardized documentation system for OD (OD-DOC), and a special documentation system for the assessment of reports related to COVID-19 (COVID-DOC). The OD-DOC data set contains the report date of the occupational disease, information about exposure, the decision, year of the decision and whether there is an entitlement to a pension due to reduced working ability (RWA). A pension is granted if the earning capacity is reduced by at least 20% for more than 26 weeks. As the examination of the claim takes time, the decision is not always made in the year in which the report is received. This results in a discrepancy between the number of decisions and reportable claims filed per year, which means that the average confirmation rate per year cannot be calculated accurately. Therefore, all claims and confirmed ODs for the investigated period were summed up and an average was calculated. Only infectious diseases transmissible from person to person were considered. Tropical infectious diseases or those transmitted by animals were excluded.

The OD-DOC allows for a distinction between the following infections: hepatitis B, hepatitis C, tuberculosis (TB) and latent tuberculosis infection (LTBI).

LTBI is present when there is a positive interferon-y release assay (IGRA) and active TB has been ruled out by X-ray. All other infectious diseases are grouped as “other infections.” The OD-DOC was analyzed for the time period from 2006 to 2019. Data from 2020 is not yet available in the OD-DOC.

The BGW operates eleven different centers nationwide that assess and document OD claims by health and social workers. At the headquarters in Hamburg, a team is responsible for the quality control of assessment and handling of data. German data protection law allows for the analysis of anonymous data from social insurance institutions including the BGW, as long as the study benefits the social security system. No written consent is needed or can be obtained from the claimant once the data set is anonymized.

## 3. Results

In the OD-DOC system, a total of 271 reportable claims of suspected ODs due to an infection among DHW were registered between 2006 and 2019 (Table 1). Most claims concerned women (88.6%), young workers (<=30 years: 36.9%) and dental assistants (76.0%). The number of claims decreased by 65.6%, from 32 claims in 2006 to 11 in 2019 (Figure 1). Of the 271 claims, a total of 112 (41.3%) were confirmed as ODs. The number of confirmed ODs decreased by 85.7%, from 14 in 2006 to 2 in 2019. For both registered claims and confirmed ODs, the trend was not monotonous but showed some variation with no apparent reason.

The number of FTE in dental medicine increased to 109.9% in 2019 compared to 2006, while the number of FTE of hospital workers increased to 195.1% during the period analyzed (Table 2). The number of claims concerning infections in hospital workers increased to 164.2% in 2016 compared to 2006, and then decreased again to 104.8% in 2019. For DHW, the number of infections decreased to 34.4% in 2019, compared to 2006. After controlling for the number of FTE per year, the rate of infections per 1000 FTE decreased from 0.89 in 2006 to 0.48 in 2019 for hospital workers (test for trend: *p* < 0.01). For DHW, the rate of infections per 1000 FTE decreased from 0.16 in 2006 to 0.05 in 2019 (test for trend: *p* < 0.001). Compared to hospital workers, the OR of infections per FTE was 0.1 (95%CI 0.06–0.19) for DHW (Table 3). Compared to general practice workers and specialist practice workers, the OR was 0.21 (95%CI 0.1–0.41) and 0.26 (95%CI 0.14–0.5) for DHW.

The decrease in infectious ODs among DHW is most evident in blood-borne infections. In the examined period, claims regarding HBV decreased by 84%, from twelve in 2006 to two in 2019. The number of confirmed ODs decreased by 75%, from four in 2006 to one in 2019 (Figure 2). There were a total of 56 claims and 33 were confirmed as OD. The average confirmation rate was 59%. HBV infections accounted for 20.6% of all claims and for 29% of all confirmed ODs. Among the 33 confirmed ODs, there are ten cases (30%) where the disease resulted in an RWA of at least 20%. The proportion of cases with RWA was relatively stable over the period studied, fluctuating between zero to three cases per year.

During the period studied, the number of confirmed occupational HCV infections decreased from ten cases in 2006 to none from 2017 onwards (Figure 3). A total of 76 cases were reported and 44 were confirmed as ODs. Among the confirmed ODs, there were 27 cases that were associated with a RWA and thus with an entitlement for pension payment. This corresponds to 61.4% of HCV-related ODs. HCV infections accounted for 28% of all claims and for 39% of all ODs, representing the largest share of ODs. The average confirmation rate was 58%.

For tuberculosis, the pattern was different. The number of confirmed ODs varied between zero and two per year with no obvious trend detectable between 2006 and 2019, while the number of claims slightly declined (Figure 4). In 2007, 2008, 2012, 2016, 2018 and 2019, the number of confirmed ODs was one or two. In the remaining years, no TB infections were confirmed as ODs. TB infections accounted for 19% of all claims (n = 52), but only for 6% of confirmed cases (n = 7), representing the smallest proportion of all ODs. The average recognition rate was the lowest, at 13%.

Claims for LTBI infections were only reported since 2010. Considering that from 2006 to 2008, no claims and confirmed ODs were registered, infection numbers increased in the period studied (Figure 5). The highest number of ODs confirmed was five in 2016; since then, the number has been decreasing. In 2019, no LTBI-related ODs were confirmed. As of 2019, a total of 36 claims were reported and 24 ODs have been confirmed, accounting for 13.3% of all claims and for 21% of all ODs. The average confirmation rate of 66.7% is the highest among all ODs.

The number of claims for other infections was 51 cases between 2006 and 2019, representing 18.8% of all claims. Most claims were registered in 2015 (n = 15), but the numbers are decreasing overall. Of the 51 claims, none were confirmed as ODs.

In the COVID-DOC reporting system of the BGW, 203 suspected occupationally acquired COVID-19 cases were registered between March 2020 and February 2021.

Of all claims, 155 (76.4%) were reportable (Table 4). For all reportable claims and four non-reportable claims, a test result was available, which was positive in 77.4% of the cases. There were 47 cases confirmed as an OD, which corresponds to a recognition rate of 38.2% of all claims with a positive test result. Most confirmed COVID-19 cases were among dental assistants and dental hygienists (76.6%), followed by dentists (21.3%). Among dental technicians, there was only one confirmed OD. Information on the course of the disease was available for 51.2% of all cases with positive test results. Of these cases, 45 (71.4%) had mild symptoms. In eleven cases (17.4%), there was a severe course. Full recovery was reported in seven cases (11.1%). No deaths due to COVID-19 infection have been reported.

The number of reportable claims for hospital workers was 28,044, or 47.3 claims per 1000 FTE. For DHW 0.7, claims per 1000 FTE were reported. The OD for DHW was 0.01 (95%CI 0.001–0.3) (no table).

## 4. Discussion

To our knowledge, this is the first study to analyze occupational infections in DHW using data from an accident insurance provider in Germany. Occupational infections are rare events in dental settings. In the OD-DOC, 112 ODs were confirmed between 2006 and 2019, representing an average of eight per year. Beginning in 2017, only two to three ODs were confirmed per year. Compared to other health workers, the risk of infection was lower and the positive time trend was even more pronounced. This shows that in relation to infectious diseases that have been confirmed and researched longer, infection prevention and hygiene standards are effective. Our data corroborates the surveys of Hübner et al., which describe improved hygiene management in dental medicine in Germany [30]. Nevertheless, DHW have an increased risk of infection in the workplace and will continue to be challenged by emerging infectious diseases such as COVID-19. In the COVID-DOC, 43 occupational COVID-19 infections were confirmed as ODs in one year only.

For the most important infections (HBV, HCV, and TB), the number of reportable infections in the general population decreased over the last 20 years in Germany. For TB, the number of cases per 100,000 population decreased from ten to five. The number of acute HBV infections decreased from more than 1000 cases in 2001 to 531 cases in 2019. The number of HCV infections decreased from 9022 cases in 2004 to 5940 cases in 2019. Therefore, due to the positive trend in the population, the decrease in the infection risk for DHW of up to 50% is likely explained. However, the claim rate per 1000 FTE was seen to have decreased from 0.16 to 0.05 (69%). This additional decrease might be explained by improved hygiene [31].

### 4.1. Hepatitis C

HCV infections accounted for the largest share of ODs, at 38%. Since 2017, no HCV infections have been confirmed by the BGW as ODs in DHW. The risk of contracting HCV occupationally is therefore very low.

Germany is a low-incidence country for hepatitis C infections, with an average prevalence of antibodies against HCV of 0.3%. However, the actual prevalence is likely to be higher, as risk groups such as HCW or drug users are usually not represented in such surveys [32]. A meta-analysis from Westermann et al. showed a significant increased prevalence of HCV infection for DHW compared with controls (odds ratio (OR) of 3.5). However, only three studies were included in this meta-analysis effect estimate, the most recent of which dates from 2005 [14].

In addition to the burden that HCV infection places on affected individuals, there are high costs associated with therapy and compensation payments. For the treatment of chronic hepatitis C, the average cost in the U.S. healthcare system is $24,176, which increases with the progress of the disease [33]. About two-thirds of the HCV infections confirmed between 2006 and 2019 at BGW resulted in an RWA. Westermann et al. examined the costs incurred by occupational HVC infections among HCWs between 1996 and 2013. Despite a decreasing number of cases, the costs associated with HVC infection have increased. During this period, pension payments due to RWA accounted for a total of 59% of expanses, and medications to treat the infection accounted for 14%. In 70% of cases with chronic hepatitis C, an initial RWA was documented [23]. Besides a small number of additional new cases every year, the increase of costs is caused by the chronic course of hepatitis C. With time, the work ability of a person with chronic hepatitis C decreases and compensation payment increases. This illustrates that hepatitis C is a disease to be taken seriously, and its transmission must be prevented. HCV infections are often asymptomatic or present with non-specific symptoms and are therefore diagnosed late. About 70% of cases take a chronic course that can lead to liver cirrhosis and hepatocellular carcinoma [34,35]. Since only reported and thus diagnosed HCV infections are recorded in our study, it is possible that the number of actual HCV infections among DHW is higher. Due to the asymptomatic course of the disease, infected patients may also not be confirmed, and their HCV infection thus not associated with occupational exposure and claimed as an OD.

There is no vaccination or post-exposure prophylaxis against HCV, so prevention in the health sector is aimed at reducing blood exposure. The U.S. Centers for Disease Control and Prevention estimate the seroconversion rate after needlestick injury with blood from an HCV-positive patient to be 1.8% [36], while a study from 2017 by Egro et al. suggests it is even lower, at 0.1% [37]. Risk of transmission depends on needle placement, depth of the injury and the viral load of the source patient [38]. Among dental assistants, most NSI occur when they are cleaning instruments (24%), followed by changing anesthetic carpules. Among dentists, most accidents happen when they are applying local anesthesia (33.3%), followed by recapping. This study shows that local anesthetic syringes pose a high risk of injury [15].

The requirements of the European Union for prevention of NSI have been implemented in Germany within the framework of “Technical Rule 250—Biological Agents in Healthcare and Welfare Facilities”. This states that the use of pointed objects should be avoided wherever possible (for example, blunt cannulas should be used for rinsing root canals). If this is not possible, safety devices that pose a low risk of needlestick injuries should be used [39]. Even though sharp safety devices (SSD) for the application of local anesthesia are available on the market, they are not yet widely used in the dental setting. Among U.S. dentists in 2008, 21% reported using SSDs in the past year, while recapping devices were used by 42% [40]. In 2018, Trayner et al. demonstrated that of 769 U.K. DHW surveyed, 49.7% used SSDs, most commonly for administration of local anesthesia [41]. It can be assumed that awareness of NSI and the resulting risks has increased, but the implementation of measures to prevent them still needs to be improved. Another measure to avoid transmission with blood-borne pathogens is to wear protective goggles, which protect the eyes from contaminated droplets. However, only 56.5% of dentists and 48.9% of assistants wear protective eyewear during every treatment [42]. It is important that DHW are regularly screened for HCV to be able to treat the disease as early as possible.

### 4.2. Hepatitis B

Hepatitis B is the second most common OD among DHW and accounted for 29% of all ODs. Since 2014, however, the infection rate has been stable, at zero to two infections per year. This decrease can be explained by a low prevalence in the general population and by the HBV vaccination. In Germany, the prevalence for acute or chronic HBV infection is 0.3% (anti-HBc positive with detection of hepatitis B virus surface antigen). 5% of the population have experienced an infection (antibodies against the core antigen of the hepatitis B virus) [32]. However, among people with a migration background who come from countries with a higher HBV prevalence, the prevalence of chronic or acute hepatitis B is significantly higher, at 3.5% [43]. Before HBV vaccination was available, the prevalence of HBV infection among dentists was estimated to be 3.5 times higher than in the general population [44]. A study in 2000 showed that among German dentists, 7% had experienced HBV infection and 1% were acutely or chronically infected [45]. For prevention of NSI, the same principles as stated above apply. Occupational HBV infections can be efficiently prevented by vaccination, which has been available since 1982. In Germany, vaccination against HBV is recommended but not mandatory [46]. Any HCW who is at increased risk of infection in the course of their employment will be offered an HBV vaccination [39]. A high vaccination rate among DHW can be assumed. In a survey of 265 DHW at the University of Frankfurt dental hospital in 2008, 88.8% reported being fully vaccinated against HBV [16]. In 2012, Ramich et al. found a 94% vaccination rate among DHW at the same university hospital [42].

### 4.3. Tuberculosis

The number of confirmed active TB infections among DHW is very low, varying from none to two cases per year. With a total of seven cases between 2006 and 2019, TB infection is the rarest OD among DHW. Germany is a low-incidence area for tuberculosis, with 5.8 new infections per 100,000 population in 2019. Two-thirds of cases were in foreign-born individuals. In the past 20 years, the incidence has dropped sharply and has almost halved compared to 2002, with a significant increase in 2015 and 2016 [47]. In the Hamburg fingerprint study, strains of M. tuberculosis complexes from 2393 TB patients were genotyped between 1997 and 2015. Among these cases were 55 HCW, of which 29 could be assigned to an infection cluster. Being an HCW was the strongest risk factor for recent transmission of TB (OR 3.07). Patient-to-HCW infection was detected in 15 cases (27.3%), which was thus the most common route of recent transmission [48]. In an epidemiological fingerprint study from the USA, it was demonstrated that 32% of 31 HCW had become infected through contact with infected patients [49].

### 4.4. Latent Tuberculosis Infection—LTBI

LTBI is the third most common OD among DHW, with a total of 24 cases.

A meta-analysis in low-incidence countries showed a pooled LTBI prevalence of 16.2% with a range from 0.9% to 85.5% for HCW in Europe [50]. Hermes et al. [51] showed a prevalence of 7.2% in HCW in general compared to 2% in workers without a connection to healthcare, corresponding to an OR of 3.86. A study among dental students in Italy found a prevalence of 2.9% when they were tested with a tuberculin skin test (TST) and/or an interferon-y release assay (IGRA) [52]. Studies among other healthcare students showed values between 0.1% and 2.1% [53,54,55,56], suggesting that dental students have a higher risk of contracting tuberculosis. The fact that occupational LTBI has only been recorded since 2009 can be attributed to a change in the perception of TB as an OD, rather than to an actual increase in infections. The notification date of the single LTBI case confirmed in 2009 is not known. The risk of infection in both HCW and DHW has long been underestimated; thus, LTBI was unlikely to have been confirmed as an OD. Epidemiologic studies revealing transmission routes and recognition of the risk of infection posed by undiagnosed patients have made it more likely that LTBI will be confirmed as an OD. Furthermore, the use of IGRAs has made diagnosis more reliable [57,58]. If LTBI is confirmed as an OD, preventive chemotherapy can be performed, which is then covered by the accident insurer. The peak in 2015 can probably be attributed to migratory movements, which are also reflected in the incidence figures of active TB [47]. Our data does not provide information on whether DHW were born in a high-incidence country or worked abroad, which is associated with a higher risk of LTBI infection. The number of LTBI is likely to be higher because infection is asymptomatic and DHW are not routinely screened for TB.

### 4.5. COVID-19

Our data shows that COVID-19 is currently the biggest health threat to DHW. Forty-seven cases of SARS-CoV-2 infection alone were confirmed as an OD by the BGW between March 2020 and February 2021. In comparison, in 2019, a total of only two ODs were confirmed, namely one HBV and one TB infection. Of the infected DHW with known disease status, 17.4% had a severe course of COVID-19. The prevalence of SARS-CoV-2 infection among DHW was estimated to be 0.8% to 2.5% in studies conducted in France [59], the United States [60] and Spain [61]. However, the studies were survey-based and the testing rate among the DHW studied was very low, ranging from 1.9% to 32.3%. In addition, these studies are prone to selection bias, as severely ill or hospitalized DHW are unlikely to be included. Furthermore, there is no information on sources of infection. A study among 499 DHW in Lombardy, Italy, conducted from May to September 2020, and aimed at quantifying SARS-CoV-2 antibody prevalence, found 10.8% of DHW to be positive [62]. Compared with a meta-analysis by Gómez-Ochoa et al., in which the pooled prevalence of SARS-CoV-2 antibodies in HCW was reported to be 7%, this rate is higher [63]. Although DHW have been identified as being at particular risk of acquiring SARS-CoV-2 infection in the workplace, data available to date shows moderate infection rates in the dental setting. Out data corroborate that infection risk in DHW is lower than in hospital workers.

These results imply that infection control in dental offices and clinics is sufficient. The low number of infections at the beginning of the pandemic may also be due to suspension of elective treatments, as many practices only offered emergency care, while some ceased all dental treatment [61,64]. A survey of German dentists in June 2020 showed that almost all participants reduced their workload, two-thirds of them by more than 50% [65]. In Italy, after the lockdown was declared, 65% of dentists reported that they performed dental emergencies or urgent care only, and 26.4% provided no dental care [8]. Another reason might be underreporting, because DHW might not have known that they treated infectious patients and therefore might not have suspected an OD.

To protect DHW from SARS-CoV-2 infection, the use of appropriate PPE is essential. In February 2020, the WHO recommended that DHW wear N95 or FFP2 respirators, gowns, gloves, eye protection and aprons when performing aerosol-generating procedures [66]. However, due to a global shortage of PPE [67,68,69,70,71], these recommendations could not be consistently implemented at the onset of the COVID-19 pandemic. Wiesmüller et al. state that, during the first wave of COVID-19 in spring 2020, only 56.7% of German dentists used FFP2 or FFP3 masks, 76.1% used protective glasses and 23.8% used gowns [72]. In March 2020, 86% of Spanish dentists reported that they had difficulties in obtaining suitable PPE, especially FFP2 masks and gowns [64]. Dental staff had significantly less access to PPE than dentists [73]. Although efforts were made internationally to increase PPE production, the shortage was still of concern to HCW in fall 2020 [71]. In addition to the use of PPE, other infection prevention measures have been implemented in many practices, such as reduced capacity in the waiting room, hand disinfection for patients and regular ventilation [72].

Since December 2020, several vaccines against SARS-CoV-2 have been licensed in the European Union [74], providing another important protective measure against COVID-19. In Germany, as in many other countries, the increased risk of infection for DHW has been confirmed, so that they are prioritized in national vaccination schedules [75,76]. It remains to be seen how high the vaccination rate among DHW will be, but there seems to be a high willingness to vaccinate. Among Greek dentists, the acceptance of vaccination is high, at 82.5% [77]. Similar results were found in a survey among Turkish HCW, in which 85% of dentists reported wanting to be vaccinated against COVID-19 [78]. Shacham et al. examined the attitudes of dentists and dental hygienists toward vaccination compared with the general population. They found significantly higher vaccine hesitancy in dental hygienists compared to dentists, while the latter did not differ significantly from the general population [79]. Kelekar et al. found that 45% of 248 dental students studied in the United States were hesitant to receive a COVID-19 vaccination. 55% of respondents said they would get vaccinated as soon as a vaccine was available, while 90% wanted more information [80]. However, these studies were conducted before or at the beginning of the national vaccination campaigns, and thus allow for only limited predictions about the actual vaccination rate.

Although COVID-19 has led to a significant increase in reported ODs, the disease is still likely to be underreported. In many cases, infection is asymptomatic or results in only mild symptoms, making it unlikely that those affected will be compensated by the BGW. This may result in infections not being reported at all.

### 4.6. Limitations

Secondary data were used for this analysis, with some limitations. Only reported cases are recorded, and limited social demographic data on insured DHW is available. For example, it is not possible to distinguish between DHW with and without immigrant backgrounds based on our data. Only DHW from non-governmental facilities are insured by BGW; employees from governmental facilities such as university hospitals with dental clinics are not included in the data.

Our data are likely subject to underreporting, since they are dependent on infections being reported. DHW with subjectively mild infections often do not find it necessary to present to a physician and thus to report the infection to accident insurance. In addition, for mild infections, there is no monetary incentive to report the infection because it is unlikely that compensation will be paid. Thus, the number of infectious ODs is likely to be higher than the available figures suggest.

A limitation of this study is that dentists are underrepresented among insured DHW at the BGW. While dental assistants, dental hygienists and dental technicians are subject to compulsory insurance with the BGW, dentists can choose their insurer. No data are available on the exact number of insured persons from the respective occupational groups. However, one advantage of secondary data is that trends can be observed over a long period of time. The data provide an overview of which occupational diseases are relevant to DHW. The inclusion of COVID-19 provides the study with additional topical relevance.

## 5. Conclusions

Infectious ODs occur very rarely in DHW. Overall, the number of confirmed ODs decreased from 2006 to 2019, which is particularly evident in blood-borne infections. Regarding older infectious diseases such as HBV, HCV and TB, universal precautions seem to be effective and to protect against transmission. The decreased prevalence in the general population in Germany also contributes to the low infection rates. Although ODs are rare among DHW, continued attention should be paid to infectious disease prevention. The greatest challenge in the dental field lies in preventing emerging infectious diseases such as COVID-19.

## Figures and Tables

**Figure 1 ijerph-18-10128-f001:**
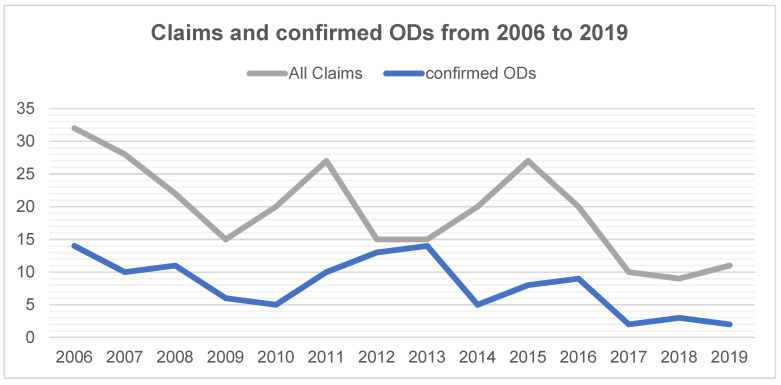
Number of claims and confirmed occupational diseases (ODs) among dental health workers between 2006 and 2019.

**Figure 2 ijerph-18-10128-f002:**
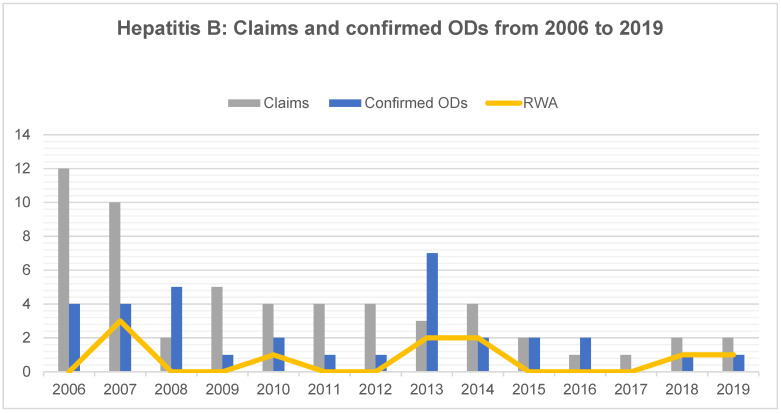
Hepatitis B virus infection: Number of claims, confirmed occupational diseases (ODs) and proportion of cases resulting in a reduced working ability (RWA) among dental health workers from 2006 to 2019.

**Figure 3 ijerph-18-10128-f003:**
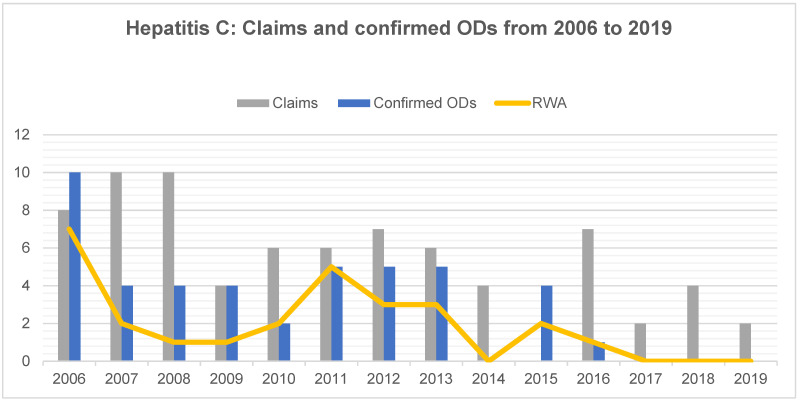
Hepatitis C virus infection: Number of claims, confirmed occupational diseases (ODs) and proportion of cases resulting in a reduced working ability (RWA) among dental health workers from 2006 to 2019.

**Figure 4 ijerph-18-10128-f004:**
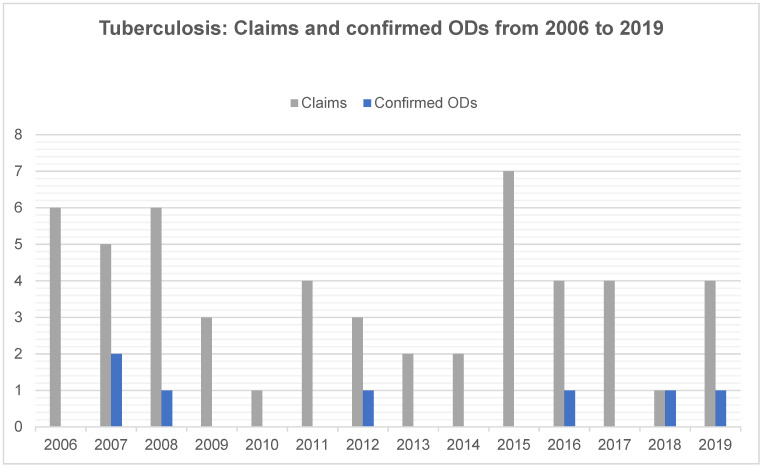
Tuberculosis: Number of claims and confirmed occupational diseases (ODs) among dental health workers from 2006 to 2019.

**Figure 5 ijerph-18-10128-f005:**
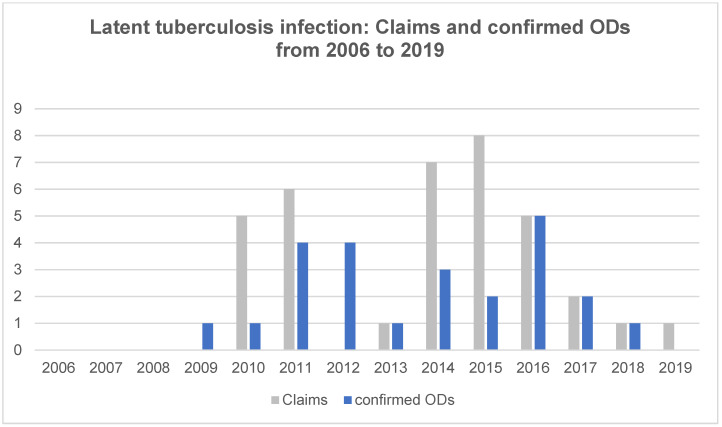
Latent tuberculosis infection: Number of claims and confirmed occupational diseases (ODs) among dental health workers from 2006 to 2019.

**Table 1 ijerph-18-10128-t001:** Description of dental health workers (DHW) filing claims due to an infection from 2006 to 2019.

Features of Population	Claims	
	N	%
Female	240	88.6
Age		
<=30 years	100	36.9
31–50 years	98	36.2
>50 years	65	24.0
Profession		
Dentist	48	17.7
Dental assistant	206	76.0
Other	17	6.3
Total	271	100.0

**Table 2 ijerph-18-10128-t002:** Full-time equivalents (FTE), claims of infection and rate of infections per 1000 FTE for 2006 to 2019 in hospital workers and dental health workers.

Year	Hospital Workers					Dental Health Workers				
	FTE in 1000		Infections		Infections/ 1000 FTE	FTE in 1000		Infections		Infections/ 1000 FTE
	N	% of 2006	N	% of 2006		N	% of 2006	N	% of 2006	
2019	592.9	195.1	284	104.8	0.48	221.0	109.9	11	34.4	0.05
2018	589.9	194.1	363	134.0	0.62	225.4	112.1	9	28.1	0.04
2017	558.2	183.6	378	139.5	0.68	224.3	111.5	10	31.3	0.04
2016	525.9	173.0	445	164.2	0.85	224.0	111.3	20	62.5	0.09
2015	507.6	167.0	318	117.3	0.63	220.3	109.5	27	84.4	0.12
2014	517.7	170.3	341	125.8	0.66	224.7	111.7	20	62.5	0.09
2013	507.9	167.1	484	178.6	0.95	223.9	111.3	15	46.9	0.07
2012	368.4	121.2	217	80.1	0.59	219.5	109.1	15	46.9	0.07
2011	356.6	117.3	258	95.2	0.72	213.4	106.1	27	84.4	0.13
2010	334.8	110.2	199	73.4	0.59	215.2	107.0	20	62.5	0.09
2009	331.9	109.2	300	110.7	0.90	213.2	106.0	15	46.9	0.07
2008	323.1	106.3	217	80.1	0.67	205.9	102.3	22	68.8	0.11
2007	318.4	104.8	236	87.1	0.74	200.2	99.5	28	87.5	0.14
2006	303.9	100.0	271	100.0	0.89	201.2	100.0	32	100.0	0.16

**Table 3 ijerph-18-10128-t003:** Non-infected full-time equivalents, number of infections by workplace and odds ratios (OD) for dental health workers (DHW) compared with hospital, general practice and specialist practice workers.

Worker	No Infection	Infection	Odds Ratio for Dental Health Workers	95%CI
Dental health	220,990	11	--	--
Hospital	592,648	284	0.1	0.06–0.19
General practice	123,432	30	0.21	0.10–0.41
Specialist practice	277,536	53	0.26	0.14–0.5

**Table 4 ijerph-18-10128-t004:** Number of claims and confirmed occupational COVID-19 infections among dental health workers from March 2020 to February 2021. OD = Occupational disease.

	Total		Dentists		Dental Assistants and Hygienists		Dental Technicians	
	N	Col%	N	Row%	N	Row%	N	Row%
Claims	203	100.0	45	22.2	152	74.9	6	3.0
Reportable claims	155	76.4	31	20.0	119	76.7	5	3.2
Cases with test result	159	78.3	34	21.4	120	75.5	5	3.1
Cases with positive result	123	77.4 ^a^	27	22.0	92	74.8	4	3.3
Confirmed OD	47	38.2 ^b^	10	21.3	36	76.6	1	2.1
Cases with known disease status	63	51.2 ^b^	16	25.4	44	69.8	3	4.8
Mild symptoms	45	71.4 ^c^	10	22.2	32	71.1	3	6.7
Severe course	11	17.4 ^c^	3	27.3	8	72.7	-	-
Full recovery	7	11.1 ^c^	3	42.9	4	57.1	-	-

^a^ of those with a test result known. ^b^ of those with a positive test result. ^c^ of those with known disease status. Col% = Column %.

## Data Availability

The data are available from the corresponding author upon request.

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
