# Peer review of "Occupational Infections among Dental Health Workers in Germany—14-Year Time Trends"

_ijerph, 2021, doi:10.3390/ijerph181910128_

Round 1
Reviewer 1 Report
Dear authors,
Thank you very much for your paper. In this paper the authors presented a study entitled: “Occupational Infections Among Dental Health Care Workers in 2 Germany – 14-Year Time Trends” to analyze time trends on occupationally acquired infections among DHCW using data from a statutory accident insurance about suspected and recognized occupational diseases (OD). The manuscript deals with an interesting issue and the topic is appropriate for the Journal. The abstract is well-written and clear. In the introduction section the issue is well addressed, but despite this, the section concerning the aerosol transmission of the virus should be implemented by considering this recent reference (see reference as Rexhepi, I.; Mangifesta, R.; Santilli, M.; Guri, S.; Di Carlo, P.; D’Addazio, G.; Caputi, S.; Sinjari, B. Effects of Natural Ventilation and Saliva Standard Ejectors during the COVID-19 Pandemic: A Quantitative Analysis of Aerosol Produced during Dental Procedures. Int. J. Environ. Res. Public Health 2021, 18, 7472. https://doi.org/10.3390/ijerph18147472 )
The discussion is very well written and should be implemented by describing also the Italian Overview during pandemic period:
Sinjari, B.; Rexhepi, I.; Santilli, M.; D′Addazio, G.; Chiacchiaretta, P.; Di Carlo, P.; Caputi, S. The Impact of COVID-19 Related Lockdown on Dental Practice in Central Italy—Outcomes of A Survey. Int. J. Environ. Res. Public Health 2020, 17, 5780. https://doi.org/10.3390/ijerph17165780
I congratulate the authors for this very relevant research, which will add to the dental field.
It appears well structured, correctly carried out and written without logical or factual errors.
Author Response
We are grateful for the thoughtful suggestions of the reviewers and we revised the article accordingly. In total we had 22 suggestions and we were able to improve the manuscript by following 22 of 22 suggestions. Please see our point-to-point response
Reviewer 1
Dear authors,
Thank you very much for your paper. In this paper the authors presented a study entitled: “Occupational Infections Among Dental Health Care Workers in 2 Germany – 14-Year Time Trends” to analyze time trends on occupationally acquired infections among DHCW using data from a statutory accident insurance about suspected and recognized occupational diseases (OD). The manuscript deals with an interesting issue and the topic is appropriate for the Journal. The abstract is well-written and clear.
1) In the introduction section the issue is well addressed, but despite this, the section concerning the aerosol transmission of the virus should be implemented by considering this recent reference (see reference as Rexhepi, I.; Mangifesta, R.; Santilli, M.; Guri, S.; Di Carlo, P.; D’Addazio, G.; Caputi, S.; Sinjari, B. Effects of Natural Ventilation and Saliva Standard Ejectors during the COVID-19 Pandemic: A Quantitative Analysis of Aerosol Produced during Dental Procedures. Int. J. Environ. Res. Public Health 2021, 18, 7472. https://doi.org/10.3390/ijerph18147472 )
Authors’ response: Thank you for pointing this out. This article was included in the introduction
2) The discussion is very well written and should be implemented by describing also the Italian Overview during pandemic period:
Sinjari, B.; Rexhepi, I.; Santilli, M.; D′Addazio, G.; Chiacchiaretta, P.; Di Carlo, P.; Caputi, S. The Impact of COVID-19 Related Lockdown on Dental Practice in Central Italy—Outcomes of A Survey. Int. J. Environ. Res. Public Health 2020, 17, 5780. https://doi.org/10.3390/ijerph17165780
Authors’ response: Thank you for the suggestion. This paper is very informative. We included this paper in the discussion.
3) I congratulate the authors for this very relevant research, which will add to the dental field.
It appears well structured, correctly carried out and written without logical or factual errors.
Authors’ response: Thank you for your positive feedback
Reviewer 2 Report
Thanks for the opportunity to review this study. Regrettably, I could not understand its scientific merit. Please consider my comments below:
-Lines 27-57
The introduction starts with diseases transmitted through air, with a focus on COVID-19 due to its recency and some concerning reports. Then it moves to the more generic cases of infections, such as hepatitis B and C. The latter concerned, the data presented are about the healthcare in overall and not the specific occupation. It does not substantiate the need to investigate dental health care staff specifically.
-Lines 80-81: The statement must be backed with literature and explain whether any general HCW studies report dental care staff in their grouping. However, a quick search showed there are several studies about diseases and infections of dental health workers. None of those is acknowledged in this paper and the gap in our knowledge is not explained and properly justified. Example papers (DOI): 10.1136/bmjopen-2016-015374, 10.1016/j.jdent.2016.04.003, 10.1136/oem.2010.057778, 10.1093/cid/ciw656, 10.1111/odi.13632, 10.3855/jidc.9911
-Methods and Results
--A merely descriptive analysis of secondary data could be useful for a technical report but could not qualify for a scientific article.
--The authors rely on raw numbers to observe a decrease in claims without considering the number of workers per period and their exposure. This data is crucial to understand any trend and the magnitude of the issue but is missing.
--Despite the comments above, since, presumably, the authors had access to all records of BGW, there is not even basic descriptive and statistical analysis against other healthcare workers and the general worker population.
Author Response
We are grateful for the thoughtful suggestions of the reviewers and we revised the article accordingly. In total we had 22 suggestions and we were able to improve the manuscript by following 22 of 22 suggestions. Please see our point-to-point response
Reviewer 2
Thanks for the opportunity to review this study. Regrettably, I could not understand its scientific merit. Please consider my comments below:
4) -Lines 27-57
The introduction starts with diseases transmitted through air, with a focus on COVID-19 due to its recency and some concerning reports. Then it moves to the more generic cases of infections, such as hepatitis B and C. The latter concerned, the data presented are about the healthcare in overall and not the specific occupation. It does not substantiate the need to investigate dental health care staff specifically.
Authors’ response: Thank you for this comment. We included additional studies from Japan, UK and US in order to show the particular concern of DHW. In addition, please consider that most of the papers we cited in this paragraph concern DHW.
5) -Lines 80-81: The statement must be backed with literature and explain whether any general HCW studies report dental care staff in their grouping. However, a quick search showed there are several studies about diseases and infections of dental health workers. None of those is acknowledged in this paper and the gap in our knowledge is not explained and properly justified. Example papers (DOI): 10.1136/bmjopen-2016-015374, 10.1016/j.jdent.2016.04.003, 10.1136/oem.2010.057778, 10.1093/cid/ciw656, 10.1111/odi.13632, 10.3855/jidc.9911
Authors’ response: Thank you for pointing this out. We agree and we rephrased the sentence. Now we included the paper on Legionella as it contained a time trend before and after 1996. In addition, we included a paper comparing hygiene management in dental practice in Germany in 2002 und 2009.
6) -Methods and Results
--A merely descriptive analysis of secondary data could be useful for a technical report but could not qualify for a scientific article.
Authors’ response: In order to improve the content of the manuscript we included comparisons with other HW and we included the number of FTE (full time equivalence) per year in dental medicine. This allows for tests of time trend and the calculation of crude odds ratios. See new table 2.
7)--The authors rely on raw numbers to observe a decrease in claims without considering the number of workers per period and their exposure. This data is crucial to understand any trend and the magnitude of the issue but is missing.
Authors’ response: We agree with the reviewer and we obtained this data. Indeed, the number of DHW increased, albeit only by 9 %, making the positive development even more pronounced. See new table 2
8)--Despite the comments above, since, presumably, the authors had access to all records of BGW, there is not even basic descriptive and statistical analysis against other healthcare workers and the general worker population.
Authors’ response: Thank you for the comment. Now we compare the number of infections as OD per 1000 FTE of dental workers, hospital workers, general practitioners, and specialized ambulatory treatment.
Reviewer 3 Report
The study is interesting. Occupational diseases must be taken into account in all professions and even more so in the health professions. However, the following questions arise and need to be addressed.
- The introduction discusses risk factors for disease in the dental practice, please note this article and add it where appropriate:
- Meethil AP, Saraswat S, Chaudhary PP, Dabdoub SM, Kumar PS. Sources of SARS-CoV-2 and Other Microorganisms in Dental Aerosols. J Dent Res. 2021 Jul;100(8):817-823. doi: 10.1177/00220345211015948. PMID: 33977764; PMCID: PMC8258727.
- In the introduction the sentence "However, studies investigating the viability of the virus under real-life conditions in a dental setting are still pending" is not supported by scientific evidence and does not add anything positive to the introduction.
- In line 57, when it talks about economic figures, in my opinion it would be correct to put: 88 million EUR.
- In the purpose of the study in line 83 to 85, it is not necessary to mention that the data have been obtained from an insurance policy, it is normally mentioned in material and methods.
- In material and method it should be improved in the following aspects:
1. It does not say what type of study it is.
2. It does not say how many operators collect the sample.
3. It should be stated whether the handling of this information has been approved by an ethics committee in accordance with the data protection law.
4. Total number of the sample, that is, each claim was a professional or includes several.
5. Define epidemiological data of the sample, sex, age, etc. If not available, include them as a limitation of the study.
- The results show the evolution or trend of a disease in relation to claims. However, it does not explain the evolution of the disease in society in those years.
- Therefore, in the discussion section, we should talk about whether the decrease of the disease in society has influenced the existence of fewer claims due to a lower probability of contagion as an occupational disease.
Table 1 shows the data from COVID-19, however, this disease is not classified as an occupational disease, but as a worldwide pandemic. As far as the diagnosis as an occupational disease is concerned, it is difficult to know whether it was contracted in the dental office or outside.
In the discussion, data from Germany are mostly used. As occupational diseases can be compared with other European countries, it would be useful to compare other studies from neighbouring countries such as France or Italy to see if the incidence has also decreased.
In your opinion, do you think it is normal for the diseases studied in this study to decrease over time or what data do you think are most relevant in your study that have not been provided by previous studies.
Thank you for de effor
Author Response
We are grateful for the thoughtful suggestions of the reviewers and we revised the article accordingly. In total we had 22 suggestions and we were able to improve the manuscript by following 22 of 22 suggestions. Please see our point-to-point response
Reviewer 3
The study is interesting. Occupational diseases must be taken into account in all professions and even more so in the health professions. However, the following questions arise and need to be addressed.
9) - The introduction discusses risk factors for disease in the dental practice, please note this article and add it where appropriate:
- Meethil AP, Saraswat S, Chaudhary PP, Dabdoub SM, Kumar PS. Sources of SARS-CoV-2 and Other Microorganisms in Dental Aerosols. J Dent Res. 2021 Jul;100(8):817-823. doi: 10.1177/00220345211015948. PMID: 33977764; PMCID: PMC8258727.
Authors’ response: Thank you for the comment. The article is included in the introduction at the end of paragraph 1.
10) - In the introduction the sentence "However, studies investigating the viability of the virus under real-life conditions in a dental setting are still pending" is not supported by scientific evidence and does not add anything positive to the introduction.
Authors’ response: Thank you for the comment. We deleted the sentence.
11)- In line 57, when it talks about economic figures, in my opinion it would be correct to put: 88 million EUR.
Authors’ response: We changed this accordingly. But the professional English lector changed it again.
12) - In the purpose of the study in line 83 to 85, it is not necessary to mention that the data have been obtained from an insurance policy, it is normally mentioned in material and methods.
Authors’ response: We agree and we made changes accordingly. The information was already given at the start of the method part and was redundant.
13)- In material and method it should be improved in the following aspects:
1. It does not say what type of study it is.
Authors’ response: Now we mention that we performed a retrospective, longitudinal study using secondary data.
14) 2 It does not say how many operators collect the sample.
Authors’ response: As we used routine data, we do not know the number of operators. Now we descript that the BGW is operating nationwide with 11 different centres assessing and documenting claims of OD of health and social workers.
15) 3. It should be stated whether the handling of this information has been approved by an ethics committee in accordance with the data protection law.
Authors’ response: As we analysed anonymous, secondary data no approval of an ethics committee was needed. Now we mention this more clearly at the end of the methods part.
16) 4. Total number of the sample, that is, each claim was a professional or includes several.
Authors’ response: It is one person one claim. Now we explain it. In case an employer reports several infections in the course of an outbreak, for every concerned worker a claim is developed.
17) 5. Define epidemiological data of the sample, sex, age, etc. If not available, include them as a limitation of the study.
Authors’ response: These data were added. See new table 1.
18) - The results show the evolution or trend of a disease in relation to claims. However, it does not explain the evolution of the disease in society in those years.
Authors’ response: Thank you for the comment. This information was added in the discussion. For TB, HBV and HCV the incidence in the general population decreased by up to 50%. Therefor part of the decrease of infections in DHW is explained by the positive development in the general population.
19)- Therefore, in the discussion section, we should talk about whether the decrease of the disease in society has influenced the existence of fewer claims due to a lower probability of contagion as an occupational disease.
Authors’ response: This discussion was added. The decrease in society alone does not explain the sharp decrease in DHCW.
20) Table 1 shows the data from COVID-19, however, this disease is not classified as an occupational disease, but as a worldwide pandemic. As far as the diagnosis as an occupational disease is concerned, it is difficult to know whether it was contracted in the dental office or outside.
Authors’ response: In Germany COVID-19 in HW, Social Workers and Laboratory Workers is considered an occupational disease. However, you are right distinguishing an infection contracted in the dental office from one contracted outside is complicated. It must be shown that a worker had contact to an infectious patient. This might be a reason why the number of COVID-19 claims for DHCW is low compared to hospital workers. We added this information and discuss it now.
21) In the discussion, data from Germany are mostly used. As occupational diseases can be compared with other European countries, it would be useful to compare other studies from neighbouring countries such as France or Italy to see if the incidence has also decreased.
Authors’ response: Thank you for this comment. We searched literature for other countries as proposed. We included a paper on legionella. However, we did not find any other publications on time trends in infections as occupational disease in DHW.
22) In your opinion, do you think it is normal for the diseases studied in this study to decrease over time or what data do you think are most relevant in your study that have not been provided by previous studies.
Authors’ response: Yes, we expected that the number of diseases declined. But we expected that the decrease would have been strong in Hepatitis B due to the vaccination and in TB due to the decrease in the German population. However, for TB we did not see such a decrease and we see a decrease in Hepatitis C, which cannot be explained by vaccination or the trend in the general population alone. See also comment 18.
Round 2
Reviewer 2 Report
Thanks for considering my comments and revising the paper. I append my remarks to this version, which I hope you find useful to increase the quality of your article.
Structure/language
-Split long paragraphs into smaller ones
-Use consistent terminology (e.g., you use confirmed and recognised ODs; pick one because the reader might get confused).
-Occasional spelling/grammar mistakes
Section 2
-There is no explanation about where the FTE figures were retrieved from. Also, it is not clear whether this is the general occupational population or only the ones insured by BGW. This must be considered when performing descriptive analysis and statistical tests and reporting the results.
-Lines 128-129: Define your α value for the tests
-Lines 130-143: Explain whether internal BGW procedures and rules have changed in the study period and how this might have affected the data you collected.
Section 3
-General comments
--Since confirmed OD trends are not comparable with claims, as explained in Section 2, what is the reason for including them in your study? What is the added value?
--Figures 2-5 can be merged into one single Table with all data.
-Line 160: Add more text about the peaks or fluctuations during this period
-As from Line 169: Explain you have defined 2006 as the year of reference.
-Lines 175-176: Report the statistical test results completely and properly.
-Table 2: Replace % of first year with % of 2006
-Add to Table 3 the p values.
-Line 208: I do not see in Figure 4 that confirmed ODs were stable. I observe another picture.
-COVID-related data were not checked for odd rations against the other categories as performed for other OD.
Section 4
-There is no need to repeat the results. This is the place to interpret them.
-When referring to OD and COVID focus on the rates per FTE and not the raw figures. For each OD type, you must discuss those figures against the prevalence of the particular OD in the German population over time (where available) and interpret accordingly. Do OD increase with OD prevalence or not and why? You have tried this for some ODs, but for the rest, a single prevalence figure does not say much.
-Several sources cited in this section are not used for any discussion of the results of the study. They are just additional information. Those should be placed in the introduction. For instance, in several instances, PPE types are mentioned. However, this cannot be linked to the results of this study as PPE were not investigated by the authors. You can place such references in the introduction and mention them briefly in this section while discussing.
-The statistical differences from other groups were not discussed (including the ones to be possibly derived from the COVID-related data as per the comment above).
-Lines 288-289: The increase in costs is not explained.
Author Response
Thanks for considering my comments and revising the paper. I append my remarks to this version, which I hope you find useful to increase the quality of your article.
Authors’ response: We thank the reviewer for taking again so much time to check our manuscript closely. Most of the suggestions were very useful and we made amendments accordingly. Change are marked in rose. The yellow marks are from the first round.
Structure/language
1) -Split long paragraphs into smaller ones
Authors’ answer: This was done, when a meaningful separation was apparent.
2) -Use consistent terminology (e.g., you use confirmed and recognised ODs; pick one because the reader might get confused).
Authors’ answer: Thank you for pointing this out. We use confirmed instead of recognised.
3) -Occasional spelling/grammar mistakes
Authors’ answer: Once again, we checked the manuscript and removed typos.
4) Section 2
-There is no explanation about where the FTE figures were retrieved. In addition, it is not clear whether this is the general occupational population or only the ones insured by BGW. This must be considered when performing descriptive analysis and statistical tests and reporting the results.
Authors’ answer: Thank you for pointing this out. Now we added: The FTE were obtained from BGW. The FTEs are an estimate of the number of workers covered by the compensation board. For example, two half-time workers add up to one FTE.
5) -Lines 128-129: Define your α value for the tests
Authors’ answer: We added that 95%CI were calculated
6) -Lines 130-143: Explain whether internal BGW procedures and rules have changed in the study period and how this might have affected the data you collected.
Authors’ answer: Now we added: During the period considered here, no change by the way of documentation or legislation concerning infections as ODs occurred.
Section 3
-General comments
7)--Since confirmed OD trends are not comparable with claims, as explained in Section 2, what is the reason for including them in your study? What is the added value?
Authors’ answer: The benefit is the estimate that the confirmation rate is high.
8) --Figures 2-5 can be merged into one single Table with all data.
Authors’ answer: Yes, you are right, but we prefer it this way.
9) -Line 160: Add more text about the peaks or fluctuations during this period
Authors’ answer: We added: For both, registered claims and confirmed ODs the trend was not monotonous but showed some variation with no apparent reason.
10) -As from Line 169: Explain you have defined 2006 as the year of reference.
Authors’ answer: Line 169 refers to the headline of the graph. We do not see how the comment on the reference year fits here. We placed it in the method part.
11)-Lines 175-176: Report the statistical test results completely and properly.
Authors’ answer: We added the 95% confidence intervals.
12)-Table 2: Replace % of first year with % of 2006
Authors’ answer: done. Thank you for pointing this out.
13) -Add to Table 3 the p values.
Authors’ answer: Adding p values is informative if the boundaries of the 95%CI are close to one. In this situation, the p-value can help to decide whether the effect is statistically significant. In table 3, no limit of the 95%CI is close to one. Therefore, we prefer not to add the p-values. Sorry for the p-value asceticism.
14) -Line 208: I do not see in Figure 4 that confirmed ODs were stable. I observe another picture.
Authors’ answer: No we say that per year between zero and two cases were confirmed. This makes it obvious that no trend can be expected with such low numbers.
15) -COVID-related data were not checked for odd rations against the other categories as performed for other OD.
Authors’ answer: The odds ratio and the 95%CI were added. It is very low 0.01.
Section 4
16) -There is no need to repeat the results. This is the place to interpret them.
Authors’ answer: Thank you for pointing this out. I think repeating the main results and commenting on them renders a discussion readable. Therefor we did not change the discussion. Please notice that two reviewers complimented us for the discussion.
17) -When referring to OD and COVID focus on the rates per FTE and not the raw figures. For each OD type, you must discuss those figures against the prevalence of the particular OD in the German population over time (where available) and interpret accordingly. Do OD increase with OD prevalence or not and why? You have tried this for some ODs, but for the rest, a single prevalence figure does not say much.
Authors’ answer: Sorry, here we disagree. Especially with the small numbers of some ODs, we prefer to discuss the absolute numbers. We discuss trends in the general population and in DHW for HB, HC, and TB because this were the main infections until the COVID-19 pandemic.
18) -Several sources cited in this section are not used for any discussion of the results of the study. They are just additional information. Those should be placed in the introduction. For instance, in several instances, PPE types are mentioned. However, this cannot be linked to the results of this study as PPE were not investigated by the authors. You can place such references in the introduction and mention them briefly in this section while discussing.
Authors’ answer: We understand your point, but again we disagree. Of cause as we have an ecologic study, we cannot refer about types of PPE. However, it is ok to discuss that the improved use of PPE might be part of the story to explain this positive trend.
19) -The statistical differences from other groups were not discussed (including the ones to be possibly derived from the COVID-related data as per the comment above).
Authors’ answer: Thank you for pointing this out. Now we added: Our data corroborate that infection risk in DHW is lower than in hospital workers.
20) -Lines 288-289: The increase in costs is not explained.
Authors’ answer: Thank you for pointing this out. Besides, of a small number of additional new cases every year, the increase is caused by the chronic course of hepatitis C. With time, workability of a person with chronic hepatitis C decreases and compensation payment increases.
Reviewer 3 Report
The article has been significantly improved. Thanks to the authors for their efforts. It is suitable for publication.
Best Regards
Author Response
Thank you for your possitive feetback.
No changes to the manuscript were requested by the reviewer.
Round 3
Reviewer 2 Report
Thanks for clarifying some of the points I commented on. You disagreed with some suggestions, but this is normal in science! I hope your work has some impact and drive policy and other initiatives.